# Polymorphism of Carbamazepine Pharmaceutical Cocrystal: Structural Analysis and Solubility Performance

**DOI:** 10.3390/pharmaceutics15061747

**Published:** 2023-06-15

**Authors:** Artem O. Surov, Ksenia V. Drozd, Anna G. Ramazanova, Andrei V. Churakov, Anna V. Vologzhanina, Elizaveta S. Kulikova, German L. Perlovich

**Affiliations:** 1G.A. Krestov Institute of Solution Chemistry RAS, Akademicheskaya Str. 1, 153045 Ivanovo, Russia; aos@isc-ras.ru (A.O.S.); ksdrozd@yandex.ru (K.V.D.); agr@isc-ras.ru (A.G.R.); 2N.S. Kurnakov Institute of General and Inorganic Chemistry RAS, Leninsky Prosp. 31, 119991 Moscow, Russia; churakov@igic.ras.ru; 3A.N. Nesmeyanov Institute of Organoelement Compounds RAS, Vavilova Str. 28, 119334 Moscow, Russia; vologzhanina@mail.ru; 4National Research Center Kurchatov Institute, 1 Kurchatova pl., 123098 Moscow, Russia; lizchkakul@mail.ru

**Keywords:** carbamazepine, methylparaben, cocrystal, polymorphism, crystal structure, stability, dissolution

## Abstract

Polymorphism is a common phenomenon among single- and multicomponent molecular crystals that has a significant impact on the contemporary drug development process. A new polymorphic form of the drug carbamazepine (CBZ) cocrystal with methylparaben (MePRB) in a 1:1 molar ratio as well as the drug’s channel-like cocrystal containing highly disordered coformer molecules have been obtained and characterized in this work using various analytical methods, including thermal analysis, Raman spectroscopy, and single-crystal and high-resolution synchrotron powder X-ray diffraction. Structural analysis of the solid forms revealed a close resemblance between novel form II and previously reported form I of the [CBZ + MePRB] (1:1) cocrystal in terms of hydrogen bond networks and overall packing arrangements. The channel-like cocrystal was found to belong to a distinct family of isostructural CBZ cocrystals with coformers of similar size and shape. Form I and form II of the 1:1 cocrystal appeared to be related by a monotropic relationship, with form II being proven to be the thermodynamically more stable phase. The dissolution performance of both polymorphs in aqueous media was significantly enhanced when compared with parent CBZ. However, considering the superior thermodynamic stability and consistent dissolution profile, the discovered form II of the [CBZ + MePRB] (1:1) cocrystal seems a more promising and reliable solid form for further pharmaceutical development.

## 1. Introduction

Carbamazepine (CBZ), commercialized under the trade name Tegretol, is a well-known anticonvulsant and neuroleptic drug that is commonly used to treat seizure disorders and neuropathic pain (Figure 1) [1]. According to the biopharmaceutical classification system (BCS), CBZ is classified as a BCS Class II drug [2] and is generally administered orally as tablets (Tegretol^®^) containing 100–400 mg of the drug. However, due to low aqueous solubility in the range of physiological pH values, CBZ exhibits inconsistent absorption, a highly fluctuating plasma concentration, and suboptimal bioavailability [3,4]. It has been reported that limited bioavailability of the drug may lead to therapeutic failure, so patients are required to take large doses to achieve the desired therapeutic effect [5,6]. On the other hand, increased blood concentrations might raise the risk of toxicity and cause severe side effects [7]. Therefore, additional formulation strategies are needed to provide the necessary release profile as well as dosage variability. A commonly employed approach that aids in the targeted refinement of a drug’s physicochemical profile is the incorporation of a pharmaceutically acceptable coformer or excipient, which modifies the packing arrangement of molecules in the solid state to produce a pharmaceutical cocrystal [8,9,10,11,12]. In fact, CBZ appeared to be one of the most intensively researched active pharmaceutical ingredients (API) in the fields of polymorphism and pharmaceutical cocrystallization. Currently, five distinct anhydrous polymorphic forms and one hydrate of CBZ have been documented in the literature [13,14,15,16]. Furthermore, the Cambridge Structural Database (CSD) [17] presently contains more than 60 instances of CBZ multicomponent crystals, including cocrystals, cocrystal polymorphs, and cocrystal solvates, and this number is growing every year. In particular, three new cocrystals of CBZ with the positional isomers of acetamidobenzoic acid have been recently described by our group [18]. Sugden et al. utilized the crystal structure prediction approach to assess the likelihood of CBZ cocrystallization with ten preselected coformers, three of which (methylparaben, 3-tert-butyl-4-hydroxyanisole and *cis*-aconitic acid) gave rise to experimentally verified novel solid forms of the drug [19]. Although multiple cocrystals of CBZ have been reported in the literature, only a handful of these solid forms contain pharmaceutically relevant or generally recognized as safe (GRAS) coformers (e.g., CBZ-nicotinamide [20,21], CBZ-saccharine [22]) and, therefore, may be deemed true pharmaceutical cocrystals and potentially utilized for further development. Therefore, an in-depth analysis of the solid form landscape and investigation of the pharmaceutically relevant physicochemical properties of the CBZ cocrystal with the GRAS coformers are critical in terms of solid dosage form formulation strategy. Since the research by Sugden et al. was limited to computational and crystallographic aspects of cocrystallization and contained no information on the physicochemical profile of the CBZ solid forms, this work inspired us to conduct further investigation of the CBZ cocrystal with the pharmaceutically relevant coformer methylparaben (Figure 1) to evaluate the impact of cocrystallization on the solubility and dissolution performance of the drug. Indeed, being a GRAS compound, methylparaben (MePRB) is rarely employed as a coformer in cocrystallization trials. It has been recently reported that cocrystal formation between the antiandrogenic compound apalutamide and MePRB improves the solubility performance and compatibility of the drug [23]. In addition, MePRB also has been utilized to obtain new pharmaceutical cocrystals and to enhance the physicochemical properties of various APIs such as propyphenazone [24], agomelatine [25], fluoroquinolone antibiotics [26,27], lamotrigine [28], quinidine [29], tadalafil [30] and ezetimibe [31]. It has also been reported that methylparaben is commonly added to the oral suspension of CBZ as a preservative compound [32], indicating that these two drugs can be co-administrated together without having a harmful effect on the pharmacokinetics or efficacy of CBZ.

In the current work, we present the results of the investigation of a novel polymorphic form of the CBZ cocrystal with MePRB in a 1:1 molar ratio as well as the channel-like cocrystal of CBZ with a calculated stoichiometry close to 1:0.25. The molecular and crystal structures of the solid forms were elucidated by X-ray diffraction methods, including single-crystal and synchrotron powder X-ray diffraction techniques, and accompanied by Raman spectroscopy. Differential scanning calorimetry (DSC), competitive slurry tests, and periodic density functional theory (DFT) calculations were also used to clarify the thermodynamic relationships between the polymorphs. Since polymorphism is known to affect the functional properties of pharmaceutical formulations [32,33,34], the dissolution behavior of polymorphic forms of the CBZ cocrystal with MePRB in aqueous media was also thoroughly investigated.

## 2. Materials and Methods

### 2.1. Compounds and Solvents

Carbamazepine (C_15_H_12_N_2_O, 98%) was received from Acros Organics (Pittsburgh, PA, USA) and identified as form III according to crystallographic parameters provided in CSD (CBMZPN01). Methylparaben (C_8_H_8_O_3_, 99%) was received from Acros Organics (Pittsburgh, PA, USA) and determined to be polymorph 1-I according to crystallographic parameters provided in CSD (CEBGOF01). Hydroxypropyl methylcellulose (HPMC, Mn~10,000) was purchased from Sigma-Aldrich (St. Louis, MO, USA). The materials were used as received. All of the solvents were of analytical or chromatographic grade.

### 2.2. Mechanochemical and Solution Crystallization

The liquid-assisted grinding tests were carried out in 12 mL agate grinding jars with ten 5 mm agate balls at a rate of 500 rpm for 60 min using a Fritsch planetary micro mill, model Pulverisette 7 (Fritsch, Idar-Oberstein, Germany). In a typical experiment, 40 μL of a solvent (acetonitrile or dichloromethane) was added with a micropipette after 80 mg of the physical mixture of CBZ and MePRB in a 1:1 molar ratio was introduced to a grinding jar. The resulting powder samples were analyzed by PXRD.

Good-quality single crystals of the CBZ cocrystal with MePRB, later identified as the channel-like cocrystal of CBZ containing a relative excess of the drug ([CBZ + MePRB] (1:0.25)), were obtained from the acetonitrile solution with a 1:1 molar ratio of the components. Single crystals of the 1:1 cocrystal could not be prepared, despite multiple trials.

The bulk powder sample of form II of the [CBZ + MePRB] (1:1) cocrystal was produced via stirring a slurry of stoichiometric amounts of the components in 0.5–0.8 mL of acetonitrile or dichloromethane for 3 days at room temperature. The resulting solid phase was separated from the solvent by vacuum filtration. The residual solvent was allowed to evaporate at room temperature and under reduced pressure for 6–8 h. Form I of the [CBZ + MePRB] (1:1) cocrystal was prepared according to the procedure described by Sugden et al., using n-heptane as a solvent [19].

### 2.3. Single Crystal XRD

The single-crystal XRD data for the [CBZ + MePRB] (1:0.25) cocrystal were collected using a Bruker D8 Venture diffractometer (Bruker AXS, Karlsruhe, Germany) with Mo-Kα radiation (λ = 0.71073 Å, graphite monochromator) at 150.0(2) K. Absorption correction was applied using SADABS [35]. The structure was solved by direct method. Non-hydrogen atoms of carbamazepine were refined in anisotropic approximation. The methylparaben molecule was highly disordered over the inversion center and glide planes, and was refined as a rigid body using AFIX 6 command in isotropic approximation for all atoms. Free refinement of occupancy gave a value close to 0.25; thus, it was fixed at this value. Positions of H(C) atoms were calculated, and those of amide and hydroxo groups were found from the difference Fourier maps. Positions of hydrogen atoms were refined isotropically in the riding model. Full-matrix least-squares refinement against F^2^ was applied using SHELXL-2018/3 [36] and OLEX2 [37] packages. Crystallographic parameters and refinement details are given in Appendix A. The CIF file is available from the Cambridge Crystallographic Data Centre under the CCDC number 2262553. This information is free of charge and can be obtained from the Cambridge Crystallographic Data Centre at www.ccdc.cam.ac.uk/structures (accessed on 1 June 2023).

### 2.4. Laboratory and High-Resolution Synchrotron Powder XRD (PXRD)

The powder XRD data for polymorph II were collected at the Kurchatov Synchrotron Radiation Source’s X-ray structural analysis beamline [38] using monochromatic radiation (λ = 0.75 Å). The sample was put in a 200 μm cryoloop and rotated across the horizontal axis throughout the test, allowing the diffraction patterns to be averaged. The patterns were acquired using a 2D Rayonix SX165 detector (Rayonix LLC., Evanston, IL, USA) placed at a distance of 250 mm with an 18° tilt angle, Debye–Scherrer (transmission) geometry, and a 400 μm beam size. The exposure time was set to 10 min. Using the Dionis software [39], the two-dimensional diffraction pattern was recorded and then integrated into the 2D plot of the intensity dependency against 2θ. To calibrate the sample–detector distance, the polycrystalline LaB_6_ (NIST SRM 660a) was used as a standard with the known position of the diffraction peaks. The diffraction peaks were approximated by the Pearson VII profile function.

The powder pattern was indexed using the Topas 5.0 software [40,41,42,43]. The structure was solved [44] and refined in the Rietveld refinement [45,46] using the Topas 5.0 as well. The solution result was used as a starting geometry for the periodic DFT calculations at the PBE-D3 level with a fixed unit cell using Quantum Espresso 7.0 [47,48]. The optimization result was further used both as the starting geometry and the source of geometry restraints in the refinement. Three isotropic thermal parameters were applied for all carbon, all oxygen and all nitrogen atoms. The positions of the hydrogen atoms were calculated and refined in the riding model. The final R value as well as the values of Pawley fit with free hkl intensities can be found in Appendix A. The difference curve was featureless, and the molecular geometry was almost unchanged after unconstrained refinement (Appendix A).

The crystallographic data for [CBZ + MePRB] (1:1) have been submitted to the Cambridge Structural Database under the CCDC number 2262554. This information is free of charge and can be obtained from the Cambridge Crystallographic Data Centre via www.ccdc.cam.ac.uk/structures (accessed on 1 June 2023).

A D2 Phaser diffractometer (Bruker AXS, Karlsruhe, Germany) equipped with a copper X-ray source (λ_CuKα1_ = 1.5406 Å) and a LYNXEYE XE-T high-resolution position-sensitive detector was applied to acquire the laboratory powder XRD data. Bragg–Brentano geometry (reflection mode) and ambient conditions were applied. The samples were loaded onto the plate sample holders and rotated while the data were being collected at a rate of 15 rpm. The PXRD patterns were recorded in the 4–30° 2θ range with a step size of 0.02° and a time/step of 1 s.

### 2.5. Raman Spectroscopy

The Raman measurements in the spectral range of 10–2000 cm^−1^ were conducted using a 3D inverted confocal Raman microscope Confotec NR500 (SOL Instruments Ltd., Minsk, Belarus) with a 50× objective lens at 532 nm excitation wavelength with a grating of 600 gr/mm. The Si wafer with the characteristic Raman line at 520 cm^−1^ was taken as a reference for calibration and basic alignment. The acquisition time and number of accumulations were optimized to increase the signal-to-noise ratio while minimizing sample deterioration. To lessen the anisotropy impact on the Raman spectra and enhance the single-to-noise ratio, all spectra for the powder samples were taken at several locations and then averaged. The acquired Raman spectra were corrected for the baseline using the asymmetric least squares smoothing method [49].

### 2.6. Thermal Analysis

A differential scanning calorimeter (DSC) with a refrigerated cooling system (Perkin Elmer DSC 4000, Waltham, MA, USA) was used for the thermal analysis of the solid forms. The sample (2–3 mg) was heated in aluminum crucibles at a rate ramp of 10 K·min^−1^ from 10 to 220 °C under a stream of dry nitrogen (20 mL·min^−1^). Indium and zinc standards were used to calibrate the device. The weighing process had an accuracy of ±0.01 mg.

### 2.7. Aqueous Solubility and Dissolution Studies

The aqueous solubility of the [CBZ + MePRB] (1:1) cocrystal form II was determined at the eutectic point, using the method described by Good and Rodriguez-Hornedo [50]. Eutectic concentrations of CBZ and MePRB were measured by suspending an excess amount of the parent CBZ (approximately 30 mg) and [CBZ + MePRB] (1:1) form II (approximately 80 mg) in 2 mL of phosphate buffer solution pH 6.5. The resulting suspension was stirred for 72 h at 37.0 ± 0.1 °C. After 72 h, a 1 mL aliquot was taken and filtered through a 0.22 μm PTFE syringe filter. The pH of the aqueous saturated solutions was measured using an FG2 pH-meter (Metler Toledo, Greifenzee, Switzerland). The equilibrium concentrations of CBZ and MePRB at the eutectic point were measured by HPLC. The solid phases after the experiment were dried at room temperature and characterized by PXRD. The result is stated as the average of three replicated experiments.

The powder dissolution experiments were carried out using an EDT-08LX dissolution tester (Electrolab, Mumbai, India), applying the USP II paddle method for 480 min. The amount of the cocrystal equivalent to 100 mg of CBZ was added to 300 mL of pH 6.5 buffer solution at 37.0 ± 0.1 °C and stirred at 75 rpm. Aliquots (approximately 1 mL) were withdrawn at predetermined time intervals (5, 10, 15, 20, 30, 45, 60, 90, 120, 180, 240, 300, 360, 420 and 480 min), and replaced with an equal volume of fresh buffer solution. The samples were then filtered and analyzed by HPLC. Dissolution tests were also carried out using a pH 6.5 buffer solution that already had 0.1% (w:v) of HPMC pre-dissolved in it. The solid residues were collected, allowed to dry at room temperature, and then further examined by PXRD following the dissolution tests. The experiments were carried out in triplicate.

### 2.8. High-Performance Liquid Chromatography (HPLC)

The concentrations of CBZ and MePRB in solution were measured using a Shimadzu LC-20 AD (Shimadzu, Kyoto, Japan) equipped with a PDA detector. A Luna C-18 column (Phenomenex, Torrance, CA, USA), 150 mm × 4.6 mm, 5 μm particle size, was used at 40 °C. A mixture of acetonitrile and a 0.1% aqueous solution of trifluoroacetic acid (35:65, v/v) was used as the mobile phase. The mobile phase flow rate was maintained at 1.0 mL·min^−1^. Carbamazepine and methylparaben were detected at λ_max_ 284 nm and 254 nm, respectively.

### 2.9. Computational Methods

The Quantum Espresso 7.0 program was used to perform the plane wave computations [47,48]. The projected augmented waves (PAW) [51,52] from PS Library version 1.0.0 and the PBE-D3 [53,54,55] and B86bPBE-XDM [56,57,58] methods were applied. For the wave functions, a kinetic energy cutoff of 60 Ry and automatic k-point sampling were used. The total energy and force convergence thresholds were set to 10^−6^ Ry and 10^−4^ Ry, respectively. The crystal structures obtained from the powder X-ray diffraction experiment served as the starting geometry for the calculations. The lattice parameters were kept constant at their crystallographic values during the geometry optimizations. The output files with the relaxed geometries were converted to cif-files using Xcrysden v.1.6.2 [59] and VESTA v.3.5.5 [60], and visualized with the Mercury software v.2022.3.0 [61].

## 3. Results and Discussion

### 3.1. Preparation and Identification of the Solid Forms

As mentioned in the Introduction section, the cocrystal between carbamazepine and methylparaben was initially identified by Sugden et al., and its crystal structure was determined from powder X-ray diffraction data [19]. This work inspired us to reproduce the solid form for further investigations, as no details of dissolution studies of the cocrystal were provided in the original publication. Apart from melt crystallization and slurry techniques suggested by Sugden et al., we also analyzed the outcomes of mechanochemical treatment of the components (in a 1:1 molar ratio) in the presence of different solvents. Surprisingly, the PXRD analysis revealed that liquid-assisted grinding of CBZ and MePRB with acetonitrile or dichloromethane resulted in the formation of a distinct product that from the reported cocrystal and parent components in powder patterns (Figure 2). PXRD revealed no evidence of the unreacted starting compounds, and the resultant solid form was thought to be a new polymorph of the [CBZ + MePRB] (1:1) cocrystal and designated as form II. The same product was obtained via a long-term slurry of the equimolar amounts of the two components in acetonitrile or dichloromethane. However, an attempt to obtain good-quality single crystals of form II through solution crystallization resulted in the formation of another distinct solid form of the cocrystal, which was contaminated by the unreacted MePRB (Figure 2), indicating that the cocrystal contains a relative excess of carbamazepine. It should be noted that the latter form was also mentioned in the Sugden et al. paper, although no structural details were provided. A number of suitable crystals of the novel form were collected from the crystallization batches and subjected to single-crystal XRD studies (Appendix A). The subsequent structural analysis revealed that the cocrystal (further referred to as [CBZ + MePRB] (1:0.25)) has a channel-like structure filled by highly disordered coformer. A similar packing framework has been reported in the literature for various CBZ cocrystals (this aspect is discussed below) [62,63,64,65]. Despite numerous attempts, we were not able to obtain the phase-pure samples of [CBZ + MePRB] (1:0.25) in a controlled manner, and only limited characterization studies were performed for this solid form.

The results of Raman spectroscopy in the mid-frequency and low-frequency areas also provided evidence for the occurrence of novel solid forms between carbamazepine and methylparaben (Figure 3). The mid-frequency spectrum of pure carbamazepine (form III) contains strong bands at 1624 cm^−1^ and 1565 cm^−1^ corresponding to non-aromatic and aromatic ν(C=C) stretching modes [66]. Another strong absorption band at 1600 cm^−1^ was assigned to the bending vibrations of the amine group δ(N-H) [66]. In the low-frequency region, the characteristic peaks of CBZ form III can be observed at 169 cm^−1^ and 183 cm^−1^ [67]. As regards the mid-frequency spectrum of parent methylparaben, the most prominent bends are located at 1675, 1589 and 1282 cm^−1^ and can be attributed to the carbonyl ν(C=O) stretching, aromatic ν(C=C) stretching and ν(C-O) vibrations, respectively [31]. The formation of a 1:1 multicomponent crystal between CBZ and MePRB causes a new vibration band to emerge in the range of 1704–1709 cm^−1^, which can be attributed to the stretching vibrations of the (C=O) of MePRB in a new crystalline environment. Most of the above-mentioned characteristic bands of the pure compounds appeared shifted in the Raman spectra of the cocrystal polymorphs, providing additional evidence for the formation of novel solid forms. Although the Raman spectra of form I and form II of [CBZ + MePRB] (1:1) in the mid-frequency range seem to be almost indistinguishable, indicating that the local environments of the molecules in these solid forms are similar, the polymorphs can be easily discriminated via the low-frequency spectral features (Figure 3) that are mainly responsible for the lattice vibrations of molecular crystals [67,68,69,70]. In contrast, the [CBZ + MePRB] (1:0.25) form exhibits a distinct spectral pattern both in mid-frequency and low-frequency regions, highlighting the fact that the molecular content and packing arrangement of this form is different from those in the [CBZ + MePRB] (1:1) polymorphs.

### 3.2. Crystal Structure Analysis and Relative Stability of the Polymorphs

The crystal structure of [CBZ + MePRB] (1:1) form II was determined from the high-resolution synchrotron powder diffraction data (Appendix A). The indexing of the unit cell dimensions and systematic absences were consistent with the P2_1_/c space group. Similarly to form I of the cocrystal described by Sugden et al. [19], [CBZ + MePRB] (1:1) form II contains one carbamazepine and one methylparaben molecule in the asymmetric unit. The components are connected via the N–H···O and O–H···O hydrogen bonds between the amide moiety of CBZ and the carbonyl oxygen and the hydroxyl group of MePRB to form a closed-ring supramolecular tetrameric unit across a crystallographic inversion center that can be described as $R_{4}^{4}(24)$ in terms of graph set notation [71,72,73] (Figure 4). In forms I and II, the spatial arrangement and geometry of the molecules involved in these tetrameric ring motifs are nearly identical (Appendix A), which explains the mentioned similarity in the mid-frequency Raman spectra of the polymorphs. In the crystal structures of both polymorphs, the four-component units are packed in distinct parallel layers, which can be envisaged along the a-axis for form I and along the b-axis for form II (Figure 4). Despite some resemblance in packing motifs between the two polymorphs, the neighboring layers in form I are found to be related by translation along the c-axis, whereas in form II, the supramolecular units in the adjacent layers are connected by the 2-fold screw axis symmetry (Figure 4). Stacks of the tetramers in polymorphs I and II are also packed in similarly; however, mutual disposition of these tectons connected by π…π or/and C-H…π bonding between CBZ molecules in two polymorphs is different (Figure 4c).

As stated above, the single-crystal X-ray analysis of the [CBZ + MePRB] (1:0.25) form revealed that the methylparaben molecules are highly disordered over the inversion center and glide planes, with the site occupation factor being close to 0.25. To further confirm the phase stoichiometry, a few single crystals of [CBZ + MePRB] (1:0.25) obtained during crystallization trials were dissolved in 1 mL of acetonitrile, and the concentrations of the constituents in the resulting solution were analyzed via HPLC using predefined calibration curves. An average molar ratio, CBZ:MePRB, was measured to be 3.5:1, which agrees well with the result of the single-crystal X-ray refinement. To gain further insight into the packing features of the [CBZ + MePRB] (1:0.25) form, we considered all of the CBZ cocrystals with >1:1 stoichiometric ratios available in the CSD [17] and analyzed their structures using the hierarchical clustering technique [62,74], performed with a Python script deposited in the official CSD GitHub repository (https://github.com/ccdc-opensource, accessed on 5 May 2023) and the CSD Python API. The resulting packing similarity dendrogram (Figure 5) showed that the obtained [CBZ + MePRB] (1:0.25) (**1**) form belongs to a cluster of isostructural crystal structures, including CBZ-4-aminobenzoic acid (**6**, 4:1, INUZAU), CBZ-4-hydroxybenzoic acid (**7**, unknown stoichiometry, MOXVIF02) and CBZ-thiourea (**14**, 2:1, UWAZID). In all of these structures, the isolated hydrogen-bonded CBZ homodimers are stacked to form cylindrical channels where disordered coformer molecules reside (Figure 6). Another distinct cluster of the isostructural channel-like structure comprises CBZ cocrystals with dicarboxylic acids (malonic acid (**8**, MOXVUR), DL-tartaric acid (**9**, MOXWIG), maleic acid (**10**, MOXWOM), and oxalic acid (**11**, MOXWUS)) and also includes the dihydrate of CBZ (**4**, FEFNOT03). As Figure 5 shows, these two clusters merge at level 10, indicating that structures belonging to different clusters have 10 molecules in common. Other systems depicted in the packing similarity tree exhibited only a minor degree of similarity with the described clusters of channel-like structures (except for **12**). For comparison, a similar procedure was applied to polymorphs I and II of [CBZ + MePRB] (1:1) and revealed clusters of 16 similarly packed molecules (Figure 4c).

To assess the relative stability of the [CBZ + MePRB] (1:1) polymorphic forms, the static crystal lattice energies, E_latt_, were calculated from periodic DFT as the difference between the total energy of the crystal divided by the number of molecules in the unit cell, Z, and the total electronic energies of the isolated molecules in the relaxed conformation. The resulting E_latt_ values showed that the new polymorph II appeared to be 1.4 kJ·mol^−1^ (B86bPBE-XDM method) more stable than the reported form I, suggesting that the packing arrangement of form II is likely to be slightly more favorable in terms of the intermolecular interactions (Appendix A). Additional evidence in favor of higher thermodynamic stability of form II was obtained from the competitive slurry experiments, which revealed that form I completely transformed to form II after slurring a mixture of polymorphs for 24 h in heptane or dichloromethane. These results clearly indicated that modification II is a thermodynamically more stable crystalline form of the [CBZ + MePRB] (1:1) cocrystal at room temperature.

### 3.3. Thermal Analysis

At the next stage, differential scanning calorimetry (DSC) experiments were performed to detect possible phase transitions between the polymorphs at elevated temperatures and evaluate the melting points of different solid phases. The DSC profile of form II recorded at a rate of 10 K·min^−1^ contains only one sharp endothermic peak (with an onset temperature of 128.9 ± 0.2 °C;), related to the cocrystal melting, while no extra endo- or exothermal events were observed (Figure 7). In contrast, two consecutive yet overlapping endothermic peaks can be seen on the DSC curve of form I, indicating that the phase transition process from form I to form II occurs prior to the melting of the cocrystal. This case of thermal behavior is ambiguous, as it is difficult to state confidently whether the polymorphs represent an enantiotropic or monotropic relationship [75]. The DSC curves recorded during the cooling process of the melted cocrystals showed no evidence of thermal events, indicating that crystallization of the cocrystal from undercooled melt is kinetically suppressed (Figure 7). The DSC experiments at higher heating rates revealed that at 20.0 K·min^−1^ and 30.0 K·min^−1^, form I melted at an onset temperature of 125.2 ± 0.5 °C; without the transformation into form II (Appendix A). According to Burger and Ramberger’s heat of fusion rule [76], a monotropic relationship between the polymorphs may be postulated since the higher melting form II has a larger enthalpy of fusion (115.0 ± 1.0 J/g) than the lower melting form I (89.0 ± 2.5 J/g). These results are in agreement with the outcome of the competitive slurry experiments and periodic DFT calculations, indicating that form II is thermodynamically more stable than form I at both room and elevated temperatures. A few crystals of the [CBZ + MePRB] (1:0.25) were subjected to DSC studies, while the TG analysis could not be performed due to an insufficient amount of the material. The resulting DSC curve had a complex shape, with the first endothermic peak occurring at 100 °C, followed by multiple thermal events that are difficult to interpret without complementary analysis (Appendix A). The obtained DSC profile revealed that [CBZ + MePRB] (1:0.25) is less thermally stable than [CBZ + MePRB] (1:1) and is likely to undergo a range of phase transitions prior to melting, indicating that the [CBZ + MePRB] (1:0.25) cocrystal is sustained by relatively weak intermolecular interactions compared to other forms. We also have to note that similar thermal behavior has been observed by Li and Matzger for the [CBZ + MePRB] (4:1) cocrystal [63], which is found to be isostructural with [CBZ + MePRB] (1:0.25).

Sugden et al. claimed that the [CBZ + MePRB] (1:1) cocrystal can also be obtained via recrystallization from the melt of a 1:1 mixture of the two components [19]. In this work, we attempted this method and performed the heat-cool-heat cycles for the physical mixture (a 1:1 molar ratio) of CBZ and MePRB using DSC (Appendix A). During the first heat, the major endothermic peak of the eutectic melting was observed at ca. 80 °C, with no sign of the exotherm that is usually attributed to the cocrystal formation. The subsequent cooling of the resulting melt down to −20 °C revealed no evidence of crystallization either, indicating the formation of a co-amorphous system. The DSC profile of the second heat of the quench-cooled material contained the glass transition event at T_g_ ≈ 11 °C (Appendix A), which is found to be significantly lower than that of pure amorphous CBZ (T_g_ = 51.1 °C) [77]. Nevertheless, the prolonged study of the quench-cooled co-amorphous form revealed that the material remained stable and showed no sign of crystallization for at least 3 days while being stored in a DSC pan at 40 °C, i.e., above T_g_. Although a detailed analysis of the obtained co-amorphous form between CBZ and MePRB is beyond the scope of the current work, the preliminary results indicate that this system is worth further exploration because of a high apparent stability rarely seen among the CBZ co-amorphous compositions [77].

### 3.4. Aqueous Solubility and Dissolution Studies of the [CBZ + MePRB] (1:1) Cocrystal Polymorphs

Solubility and dissolution performance are of particular interest due to their relationship with drug absorption and bioavailability. The cocrystal solubility advantage (SA) over the drug is known to be related to the coformer solubility in aqueous media [78]. Given that MePRB (2.55·10^−2^ M at 37 °C [79]) is 31 times more soluble in water than CBZ dihydrate ((8.30 ± 0.02)·10^−4^ M at 37 °C [18]), the combination of these substances is expected to produce a more soluble cocrystal than the parent crystalline API.

Since form II of [CBZ + MePRB] (1:1) was found to be a more stable solid form at room temperature, the thermodynamic solubility of this polymorphic form was evaluated. The preliminary stability experiments showed that form II dissolves incongruently and undergoes a phase transformation to CBZ dihydrate in the pH 6.5 buffer solution at 37 °C. Therefore, the thermodynamic solubility of the cocrystal was assessed by measuring the CBZ and MePRB concentrations at the eutectic point [50], where the cocrystal and CBZ dihydrate are in equilibrium with a solution (Appendix A). The experimental values for the CBZ and MePRB eutectic concentrations, as well as the calculated value of the eutectic constant, are summarized in Appendix A. The eutectic constant was calculated to be greater than 1, which indicates that [CBZ + MePRB] (1:1) form II is more soluble than the parent CBZ in a given medium. The resulting value for the thermodynamic solubility of the cocrystal in pH 6.5 buffer solution and at 37.0 °C was 2.96 ± 0.04·10^−3^ M. Considering the CBZ dihydrate solubility under the same conditions, the cocrystal SA parameter equaled 3.6 ± 0.1. This value represents the maximum theoretical supersaturation that can be generated by the [CBZ + MePRB] (1:1) cocrystal during the dissolution process in the aqueous solution. It is well known, however, that the cocrystal solubility advantage could be compromised due to rapid solution-mediated phase transformation (SMPT) and precipitation of a less soluble solid form, as observed for other cocrystals [80,81,82].

The ability of the studied cocrystals to generate and maintain supersaturation during the dissolution process was monitored under non-sink conditions in a pH 6.5 buffer solution at 37 °C (Figure 8). As Figure 8 illustrates, the polymorphs of [CBZ + MePRB] (1:1) showed similar dissolution profiles, whereas the parent CBZ was characterized by a much slower dissolution. The anhydrous form of CBZ is known to undergo a phase transformation to the dihydrate form in an aqueous medium. However, after 480 min of CBZ dissolution, the drug concentration was significantly lower than the equilibrium solubility of CBZ dihydrate ((8.30 ± 0.02)·10^−4^ M) due to the low dissolution rate of the compound. The PXRD analysis of the samples recovered after the dissolution experiment confirmed the partial phase conversion of the anhydrous CBZ to its dihydrate form (Appendix A). Although the [CBZ + MePRB] (1:1) cocrystal was confirmed to be a more soluble than the parent drug, supersaturation was not observed during the dissolution test, presumably because of the rapid SMPT process and precipitation of CBZ dihydrate. The phase transformation of the cocrystal polymorphs to CBZ dihydrate was additionally verified by PXRD analysis of the residual solids after the dissolution experiments (Appendix A). Both polymorphs of [CBZ + MePRB] (1:1), however, exhibited a high dissolution rate, reaching a plateau value equivalent to CBZ dihydrate equilibrium solubility within a few hours of the dissolution test. The reasons behind the improved dissolution rate could be attributed to the high solubility of MePRB with respect to CBZ and rapid cocrystal dissociation during the dissolution. Similar behavior has been reported in the literature for other CBZ cocrystals, and the use of pharmaceutically acceptable polymers has been proven to be an effective strategy for inhibiting the fast SMPT of cocrystals [81,83,84]. In the present study, the HPMC excipient was employed owing to its ability to prevent CBZ from precipitating in the bulk phase within a few hours [84]. As Figure 8 shows, the presence of pre-dissolved HPMC (0.1% w/v) in the dissolution medium had a significant effect on the dissolution behavior of both polymorphs of the [CBZ + MePRB] (1:1) cocrystal. Considering the CBZ and MePRB concentration ratio during the dissolution experiment, [CBZ + MePRB] (1:1) form I exhibited incongruent dissolution, suggesting the cocrystal dissociation in the solid phase (Appendix A). The dissolution profile of [CBZ + MePRB] (1:1) form I demonstrated a rapid increase in the CBZ concentration within the first 60 min of the experiment, reaching a plateau level at 120 min. The CBZ content at the end of the experiment was ca. 1.6 times higher than that of CBZ dihydrate. Interestingly, the PXRD analysis of the residual solid revealed the presence of anhydrous CBZ, while only traces of CBZ dihydrate were detected (Appendix A). This is likely due to the reported ability of HPMC to inhibit the phase transformation of more soluble anhydrous CBZ to the dihydrate form [85,86]. In contrast, the dissolution of [CBZ + MePRB] form II in buffer solution with pre-dissolved HPMC was characterized by a congruent dissolution with a steady increase in the CBZ and MePRB concentration throughout the experiment (Appendix A). Although the CBZ concentration values for both cocrystal polymorphs at the end of 480 min were comparable, it is clear that the drug concentration for the [CBZ + MePRB] form II had not reached a plateau level. The residual solid recovered after the cocrystal dissolution experiment was identified as [CBZ + MePRB] (1:1) form II by PXRD (Appendix A). These results indicate that the addition of HPMC seems to completely inhibit the SMPT of the cocrystal to the parent drug for a relatively long time period, presenting a potential improvement in bioavailability.

## 4. Conclusions

Polymorphism plays a vital role in contemporary drug development, and investigations of this phenomenon in the context of the solid-state properties of single- and multicomponent crystals are of great importance for both academia and the pharmaceutical industry. In the current publication, a novel polymorphic form of the pharmaceutical cocrystal between carbamazepine and methylparaben in a 1:1 molar ratio ([CBZ + MePRB] (1:1) form II) as well as the channel-like cocrystal of the drug containing highly disordered coformer molecules with a molar stoichiometry of 1:0.25 ([CBZ + MePRB] (1:0.25)) were reported. The resulting solid forms were fully characterized using a variety of solid-state characterization methods, including single-crystal and high-resolution synchrotron powder X-ray diffraction, Raman spectroscopy, and thermal analysis. The first polymorph (form I) of [CBZ + MePRB] (1:1) was initially obtained and described by Sugden et al. [19]. According to the results of structural analysis, the crystal structures of the [CBZ + MePRB] (1:1) polymorphs appeared to be similar in terms of overall packing arrangements and stabilized by the same robust $R_{4}^{4}(24)$ heterosynthon of the hydrogen bonds formed by the amide moiety of CBZ and the carbonyl oxygen and the hydroxyl group of MePRB. In turn, the [CBZ + MePRB] (1:0.25) cocrystal was found to belong to a distinct family of isostructural crystal structures, where the isolated hydrogen-bonded CBZ homodimers are stacked to form confined channels filled by disordered coformer molecules. According to the DSC studies and competitive slurry experiments, forms I and II of the 1:1 cocrystal are monotropically related, with novel form II being confirmed as the thermodynamically more stable solid phase. In addition, the aqueous dissolution studies revealed that moderate differences in the packing arrangements of the polymorphs have a great impact on their dissolution behavior, especially in the presence of the precipitation inhibitor. It was found that utilizing the HPMC polymer as an excipient enabled the suppression of the nucleation and precipitation of CBZ during the dissolution of form II, resulting in the congruent dissolution of the cocrystal. In contrast, solution-mediated phase transformation and incongruent dissolution were observed for the metastable and more soluble form I. Considering the superior thermodynamic stability and consistent dissolution profile, the described form II of the [CBZ + MePRB] (1:1) cocrystal seems a more reliable and promising crystalline form to use in subsequent stages of pharmaceutical development.

## Figures and Tables

**Figure 1 pharmaceutics-15-01747-f001:**
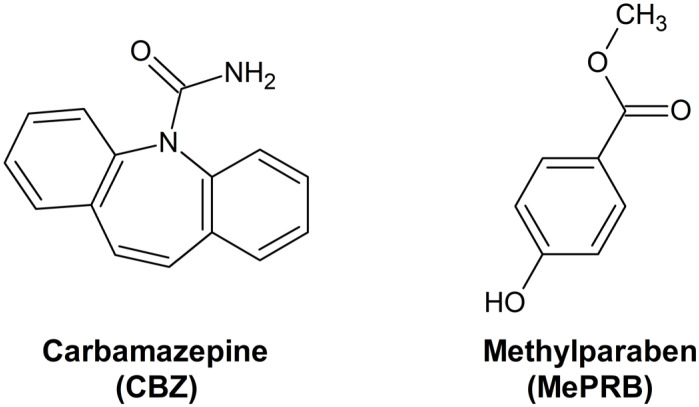
Molecular structures of carbamazepine and methylparaben.

**Figure 2 pharmaceutics-15-01747-f002:**
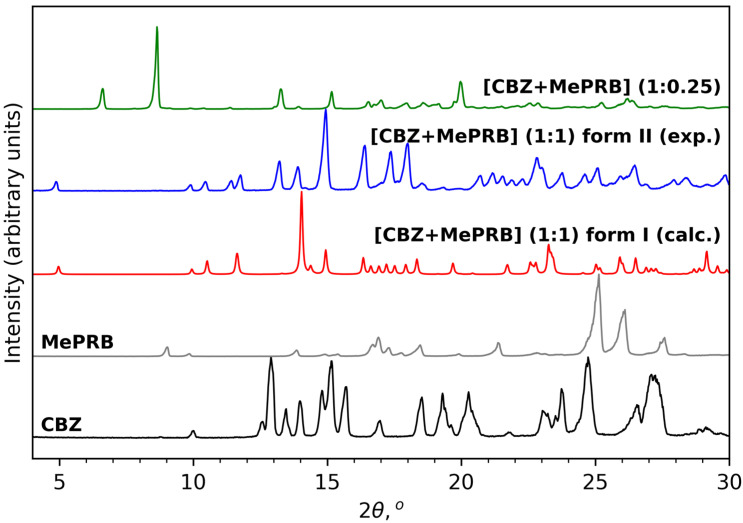
PXRD patterns of CBZ (P2_1_/c), MePRB (Cc), [CBZ + MePRB] (1:1) form I (Pī) and form II (P2_1_/c), and [CBZ + MePRB] (1:0.25) channel-like cocrystal (C2/c).

**Figure 3 pharmaceutics-15-01747-f003:**
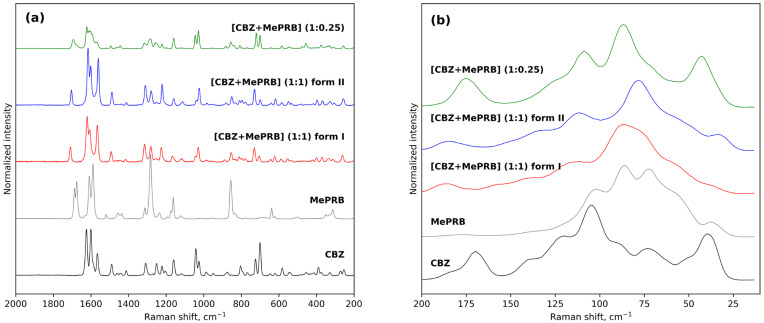
Experimental Raman spectra of CBZ, MePRB, [CBZ + MePRB] (1:1) form I and form II, and [CBZ + MePRB] (1:0.25) channel-like cocrystal in the mid-frequency (**a**) and low-frequency spectral (**b**) regions.

**Figure 4 pharmaceutics-15-01747-f004:**
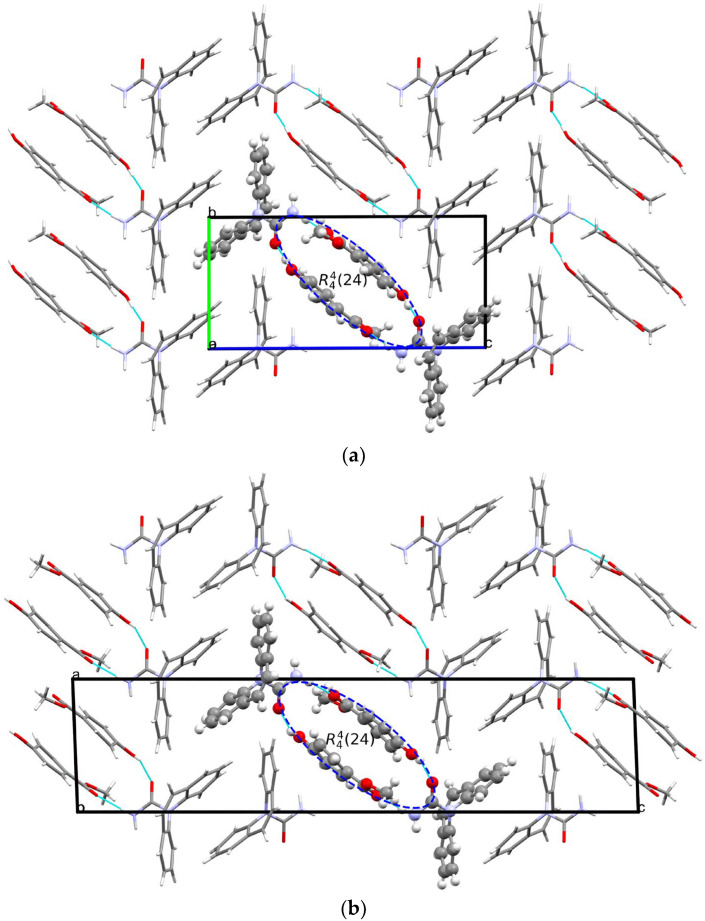
Illustration of hydrogen bonding motifs and molecular packing arrangements in the crystal structures of form I (**a**) and form II (**b**) of [CBZ + MePRB] (1:1). Color code: C, grey; H, white; N, blue; O, red. (**c**) Comparison of two polymorphs depicted with overlaid motifs (polymorphs I and II are colored with green and red).

**Figure 5 pharmaceutics-15-01747-f005:**
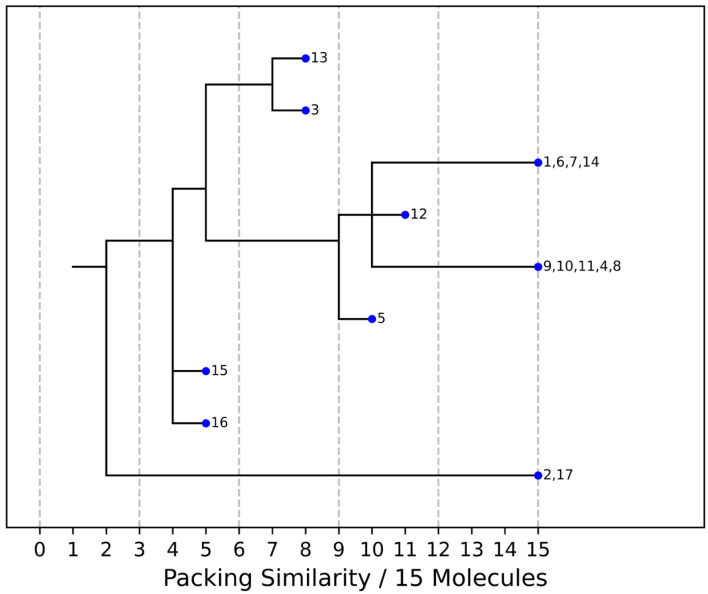
Packing similarity dendrogram for CBZ multicomponent crystals with >1:1 stoichiometric ratio. Structures grouped (e.g., **1**, **6**, **7**, **14**) on the right-hand side indicate that 15 out of 15 molecules in a cluster are similar, and these are considered to be isostructural. **1**—[CBZ + MePRB] (1:0.25), **2**—CBZ-4-aminosalicylic acid hydrate (2:1:1, FAYXUB), **3**—CBZ-4-aminosalicylic acid methanol solvate (2:1:1, FAYYAI), **4**—CBZ dihydrate (FEFNOT03), **5**—CBZ furfural solvate (2:1, FOMXAH), **6**—CBZ-4-aminobenzoic acid (4:1, INUZAU), 7—CBZ-4-hydroxybenzoic acid (MOXVIF02), **8**—CBZ-malonic acid (MOXVUR), **9**—CBZ-DL-tartaric acid (MOXWIG), **10**—CBZ-maleic acid (MOXWOM), **11**—CBZ-oxalic acid (MOXWUS), **12**—CBZ-1,4-dioxane solvate (2:1, QABHOU01), **13**—CBZ-anthranilic acid (2:1, RUTGOE), **14**—CBZ-thiourea (2:1, UWAZID), **15**—CBZ-4,4′-bipyridine (2:1, XAQQUC), **16**—CBZ-4-aminobenzoic acid (2:1, XAQRAJ), **17**—CBZ-4-aminobenzoic acid hydrate (2:1:1, XAQREN).

**Figure 6 pharmaceutics-15-01747-f006:**
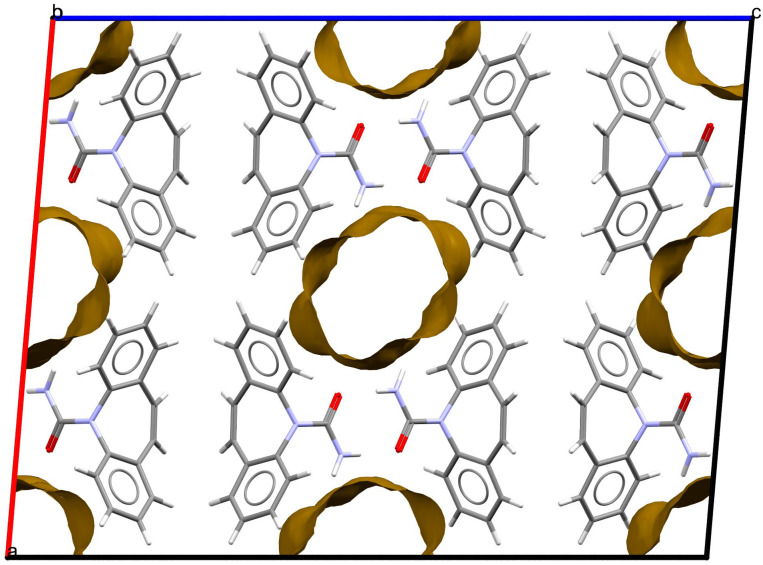
Fragment of packing arrangement and void maps in the crystal structures of the [CBZ + MePRB] (1:0.25) form. The disordered molecules of methylparaben are not shown. Color code: C, grey; H, white; N, blue; O, red.

**Figure 7 pharmaceutics-15-01747-f007:**
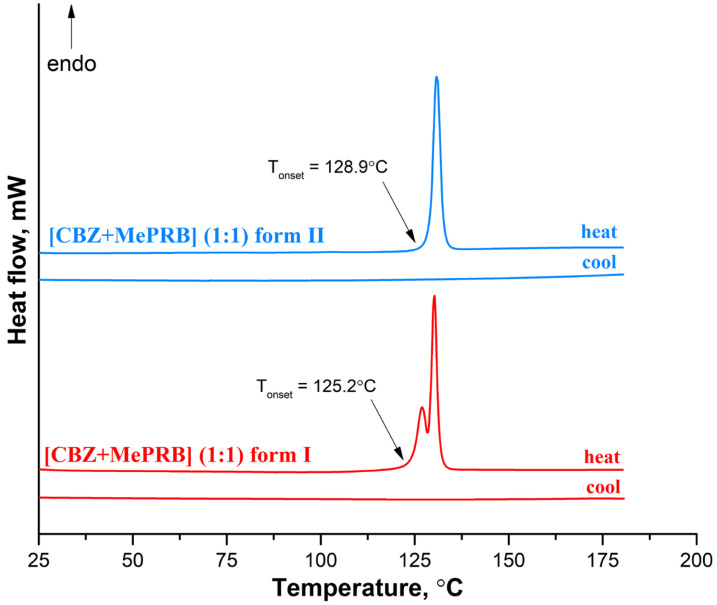
DSC curves for form I and form II of the [CBZ + MePRB] (1:1) cocrystal recorded at a heating rate of 10 K·min^−1^.

**Figure 8 pharmaceutics-15-01747-f008:**
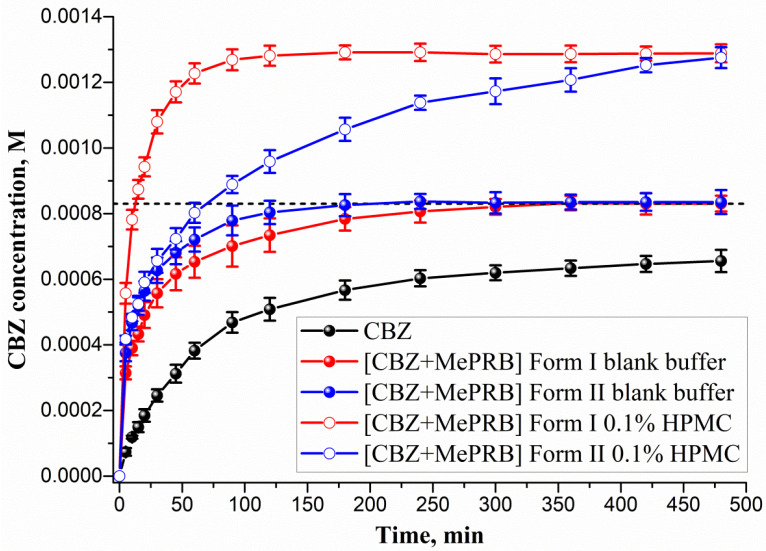
Powder dissolution profiles in pH 6.5 aqueous solution at 37 °C for CBZ, [CBZ + MePRB] (1:1) form I and [CBZ + MePRB] (1:1) form II with and without pre-dissolved HPMC polymer (0.1% w/v). The dash lines correspond to the CBZ dihydrate solubility.

## Data Availability

The results obtained for all experiments performed are shown in the manuscript and Appendix A; the raw data will be provided upon request.

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
