# Peer review of "Polymorphism of Carbamazepine Pharmaceutical Cocrystal: Structural Analysis and Solubility Performance"

_pharmaceutics, 2023, doi:10.3390/pharmaceutics15061747_

Round 1
Reviewer 1 Report
In this study, authors reported the polymorphism of carbamazepine cocrystal with methylparaben. Various solid-state characterization techniques were applied to characterize the solid forms and at the same time, theoretical calculations were carried out to determine and analyze the structures. This is a systematic and scientific study. However, this manuscript could be accepted after some problems are addressed.
1. The crystal structure file of 2262553 was not provided by the authors and it is also not available on the CCDC website. Therefore, I cannot judge whether the author's analysis of the structure is comprehensive or not.
2. Please add DSC and TG experiments for the cocrystal (1:n) to obtain the corresponding thermodynamic information.
3. Is it possible for the cocrystal (1:n) to have crystalline solvents or water?
4. In Figure 2 and S5, the PXRD profile of form II seemed to be the same, but were labeled by “exp” and “cal”, respectively. Please check.
Author Response
In this study, authors reported the polymorphism of carbamazepine cocrystal with methylparaben. Various solid-state characterization techniques were applied to characterize the solid forms and at the same time, theoretical calculations were carried out to determine and analyze the structures. This is a systematic and scientific study. However, this manuscript could be accepted after some problems are addressed.
Comment:
- The crystal structure file of 2262553 was not provided by the authors and it is also not available on the CCDC website. Therefore, I cannot judge whether the author's analysis of the structure is comprehensive or not.
Response:
We re-refined the structure of the cocrystal with methylparaben as a rigid body. Free refinement of its occupancy gives value close to 0.25, that is the maximum of theoretically possible for this crystallographic position. Thus, occupancy was fixed at 0.25. CIF file was resubmitted to the CSD and attached to this manuscript.
Comment:
- Please add DSC and TG experiments for the cocrystal (1:n) to obtain the corresponding thermodynamic information.
Response:
A few crystals of the [CBZ+MePRB] (1:n) were subjected to DSC studies, while the TG analysis could not be performed due to an insufficient amount of the material. The resulting DSC curve had a complex shape, with the first endothermic peak occurring at 100°C, followed by multiple thermal events that are difficult to interpret without complementary analysis (Figure S4). The obtained DSC profile revealed that [CBZ+MePRB] (1:n) has weaker thermal stability than [CBZ+MePRB] (1:1) and is likely to undergo a phase conversion before reaching its melting point, which is suggestive of relatively weak intermolecular interactions in the [CBZ+MePRB] (1:n) cocrystal compared to other forms. We also have to note that similar thermal behavior has been observed by Li and Matzger for the [CBZ+MePRB] (4:1) cocrystal (Li, Z., & Matzger, A. J. (2016). Molecular pharmaceutics, 13(3), 990-995), which is found to be isostructural with [CBZ+MePRB] (1:n). The DSC results of the [CBZ+MePRB] (1:n) form were added to the supporting information (Figure S4). The relevant discussion was introduced in Section 3.3, Thermal Analysis.
Comment:
- Is it possible for the cocrystal (1:n) to have crystalline solvents or water?
Response:
We re-refined the structure of the cocrystal with methylparaben as a rigid body. Some residual density is still present in the channel; however, it is impossible to assign it to either thermal motion of a disordered methylparaben, or some water molecules. As mentioned above, the DSC measurement of the [CBZ+MePRB] (1:n) cocrystal showed no thermal event below 100C, suggesting the presence of no or a minimal quantity of a solvent in the crystal.
Comment:
- In Figure 2 and S5, the PXRD profile of form IIseemed to be the same, but were labeled by “exp” and “cal”, respectively. Please check.
Response:
We thank the reviewer for careful examination of the manuscript. The label in Figure S6 was corrected.

Reviewer 2 Report
This paper represents a very interesting and comprehensive investigation on the carbamazepine-methylparaben cocrystal. The authors have found a second 1:1 polymorph of this system that, despite presenting a very similar crystallographic structure compared to the previously reported phase I, has significantly improved solubility performance.
The work is indeed very interesting and thorough and, with the exception of some very minor typing mistakes - see, for example, line 56, line 98 ("from the" instead of "from a"), line 201, ... - the manuscript is ready to be published as it is.
Author Response
This paper represents a very interesting and comprehensive investigation on the carbamazepine-methylparaben cocrystal. The authors have found a second 1:1 polymorph of this system that, despite presenting a very similar crystallographic structure compared to the previously reported phase I, has significantly improved solubility performance.
Comment:
The work is indeed very interesting and thorough and, with the exception of some very minor typing mistakes - see, for example, line 56, line 98 ("from the" instead of "from a"), line 201, ... - the manuscript is ready to be published as it is.
Response:
We would like to thank the Reviewer for his/her careful inspection of the manuscript and kind evaluation of its content. The typing mistakes were corrected throughout the manuscript.

Reviewer 3 Report
The aim of the manuscript is to describe the cocrystal polymorphism of carbamazepine.
Carbamazepine is an old and well-known drug. There are many solved structures in the CSD. In this manuscript, no problem of administering this drug or description of the need for another solid form is outlined. On the other hand, the crystallographic study regarding the "channel" cocrystal is interesting.
The novelty of this paper in the field of drugs is not sufficient for the journal Pharmaceutics, so I suggest that this paper be submitted to Crsytals.
Author Response
The aim of the manuscript is to describe the cocrystal polymorphism of carbamazepine.
Carbamazepine is an old and well-known drug. There are many solved structures in the CSD. In this manuscript, no problem of administering this drug or description of the need for another solid form is outlined. On the other hand, the crystallographic study regarding the "channel" cocrystal is interesting.
The novelty of this paper in the field of drugs is not sufficient for the journal Pharmaceutics, so I suggest that this paper be submitted to Crsytals.
We have to stress that the manuscript has been deliberately designed for the special issue "Polymorphism" of the Pharmaceutics journal (https://www.mdpi.com/journal/pharmaceutics/special_issues/5IDXR9GWAC), and its content meets all the bullet points provided in the description of the special issue. In particular, “Screening the polymorphic forms of drug compounds and multicomponent molecular crystals (drug–drug, drug–coformer) by different methods and approaches…”, “Analysis of the crystal structures of polymorphic pharmaceutical systems…”, “Theoretical calculations/evaluations (by various approaches) of the crystal lattice energies of different polymorphic forms…”, “In vivo and in vitro studies of polymorphic modifications”. Therefore, the manuscript mainly focuses on structural analysis, the relative stability of the polymorphs, and differences in their physicochemical properties, as the listed features are highly relevant in this particular case and are likely to be of interest to potential readers of the special issue.
Furthermore, as mentioned in the Introduction section, carbamazepine is indeed one of the most thoroughly investigated drug compounds in the realm of cocrystallization and crystal engineering, with over 60 entries currently present in the Cambridge Structural Database (CSD). However, only a few of the reported solid forms contain GRAS coformers (i.e., pharmaceutically acceptable additives) and, therefore, may be deemed true pharmaceutical cocrystals and potentially utilized for further development. Since the carbamazepine cocrystal reported in this work belongs to a short list of pharmaceutical cocrystals of the drug, as it is formed with the GRAS compound methylparaben, it should not be treated as a conventional cocrystal with a non-GRAS coformer, and it apparently deserves an in-depth analysis of its solid form landscape and pharmaceutically relevant physicochemical properties.
This study also highlights the problem of the so-called late-appearing polymorphism, i.e., the phenomenon when a previously unknown and more stable polymorphic form of a drug compound crystallizes, even though the known solid form has been considered monomorphic [Bučar, D. K. et al. (2015) Angewandte Chemie International Edition, 54(24), 6972-6993; Mortazavi, M., et al. (2019). Communications Chemistry, 2(1), 70; Taylor, C. R. et al. (2020). Journal of the American Chemical Society, 142(39), 16668-16680]. It is known that a late-appearing polymorphism can have significant implications for the efficacy and safety of a drug, as the newly discovered form may have different physicochemical and pharmacokinetic profiles. To mitigate the risk of late-appearing polymorphism occurrence during the early stages of drug development, a polymorph screening and characterization of alternative solid phases, such as that performed in this work, are required.
Finally, although carbamazepine is a relatively old drug, its polymorphic behavior and multicomponent solid forms are still being extensively studied. For example, a comprehensive study of the solid-state landscape of carbamazepine during its dehydration has been very recently published in Pharmaceutics [Remoto, P. I. J et al. (2023). Pharmaceutics, 15(5), 1526]. Earlier this year, we also reported the structural analysis and thermodynamics for new cocrystals of the drug with the positional isomers of acetamidobenzoic acid [Surov, A. O. et al. (2023). Pharmaceutics, 15(3), 836.]. In addition, two papers concerning structural aspects and disorder of carbamazepine with dl-tartaric acid and l-tartaric acid (l-TA) have been recently presented by Roca-Paixão et al. [Roca-Paixão, L. et al. (2022). Cryst. Growth Des., 23, 3, 1355–1369; Roca-Paixão, L. et al. (2023) Cryst. Growth Des., 23, 1, 120–133]
Overall, the current study emphasizes the importance of understanding the polymorphism of pharmaceutically relevant solid forms, such as carbamazepine cocrystal with methylparaben, in order to mitigate the consequences of an unwanted alteration of the physicochemical properties of a formulation during various development stages. Therefore, it seems reasonable to assert that the manuscript is suited to the Pharmaceutics journal in terms of subject areas and scopes. The Introduction section has been amended to include several sentences highlighting the importance of developing novel soluble dosage forms of carbamazepine via the cocrystallization technique, as suggested by the reviewer.

Reviewer 4 Report
Artem O. Surov and other coworkers presented a thorough report on a new polymorphism form of carbamazepine (CBZ) cocrystal with methylparaben (MePRB). They have done detailed structural analysis using various techniques and solubility performance and potential application in pharmaceutical development. They have compared their analysis with a previously reported complex.
The research work is informative and interesting.
I would like to suggest the authors make some changes. I recommend the manuscript be published in Pharmaceutics after that.
(1) The authors have indicated that [CBZ+MePRB] (1:1) cocrystal is more promising and reliable solid form for further pharmaceutical development. I think it is important to describe in detail what kind of pharmaceutical development the authors are expecting. Please give some examples. Then the whole article will be more apt for the readerships of Pharmaceutics journal.
(2) There are a lot of mentions of [CBZ+MePRB] (1:n). Please mention the value of n.
(3) In the powder XRD analysis (Figure 2), please mention the lattice if possible.
(4) Please mention the color codes of different atoms used for the molecular models in Figure 4 and in Figure 6.
(5) There is a small disruption in the DSC peak for [CBZ+MePRB] (1:1) (Form I). What does that correspond to (Figure 7)? Some readers may be interested to know how the DSC traces look like while cooling (identical or not).
Author Response
Artem O. Surov and other coworkers presented a thorough report on a new polymorphism form of carbamazepine (CBZ) cocrystal with methylparaben (MePRB). They have done detailed structural analysis using various techniques and solubility performance and potential application in pharmaceutical development. They have compared their analysis with a previously reported complex.
The research work is informative and interesting.
I would like to suggest the authors make some changes. I recommend the manuscript be published in Pharmaceutics after that.
Comment:
(1) The authors have indicated that [CBZ+MePRB] (1:1) cocrystal is more promising and reliable solid form for further pharmaceutical development. I think it is important to describe in detail what kind of pharmaceutical development the authors are expecting. Please give some examples. Then the whole article will be more apt for the readerships of Pharmaceutics journal.
Response:
The Introduction section has been amended to include several points highlighting the importance of developing novel soluble dosage forms of carbamazepine via the cocrystallization technique, as suggested by the reviewer.
Comment:
(2) There are a lot of mentions of [CBZ+MePRB] (1:n). Please mention the value of n.
Response:
The crystal structure of this form was re-refined. The occupancy of methylparaben was obtained as 0.25. (1:n) was replaced by (1:0.25) throughout the text. The necessary corrections were introduced in the main text and supporting information.
Comment:
(3) In the powder XRD analysis (Figure 2), please mention the lattice if possible.
Response:
The space group notation for each solid was added to the figure caption.
Comment:
(4) Please mention the color codes of different atoms used for the molecular models in Figure 4 and in Figure 6.
Response:
The legend for atom types was added to the captions of Figures 4 and 6. For Figures 4 and 6 the color code is: C, grey; H, wight; N, blue; O, red.
Comment:
(5) There is a small disruption in the DSC peak for [CBZ+MePRB] (1:1) (Form I). What does that correspond to (Figure 7)? Some readers may be interested to know how the DSC traces look like while cooling (identical or not)
Response:
As mentioned in the main text, the first low-temperature peak on the DSC curve of form I (with an onset temperature of 125.2C) should be attributed to the melting process of form I, which is followed by phase transition to form II and subsequent melting of the latter form.
As suggested, the DSC curves recorded during the cooling process of the melted cocrystals were added to Figure 7. However, no evidence of thermal events was detected, indicating that crystallization from undercooled melt is kinetically suppressed. A similar behavior was observed during the heat-cool-heat cycles for the physical mixture of the components (a 1:1 molar ratio), as described in the main text. The necessary corrections were introduced in the main text.

Round 2
Reviewer 3 Report
The aim of the manuscript is to describe the polymorphism of carbamazepine cocrystals.
The novelty of this article makes it more suitable for Crsytals than Pharmaceutics, however, the extended introduction helps to make it acceptable for a special issue of Pharmaceutics.